# Immunoreactive Trypsinogen and Free Carnitine Changes on Newborn Screening after Birth in Patients Who Develop Type 1 Diabetes

**DOI:** 10.3390/nu13103669

**Published:** 2021-10-19

**Authors:** Jane Frances Grace Lustre Estrella, Veronica C. Wiley, David Simmons

**Affiliations:** 1Macarthur Clinical School, Western Sydney University School of Medicine, Campbelltown 2560, Australia; J.Estrella2@westernsydney.edu.au; 2NSW Newborn Screening Program, Locked Bag 2012 Wentworthville, Sydney 2145, Australia; veronica.wiley@health.nsw.gov.au; 3Genetic Medicine, Discipline of Paediatrics and Child Health, University of Sydney, Sydney 2006, Australia

**Keywords:** newborn screening, dried blood spots, type 1 diabetes, acylcarnitines, metabolic profile, carnitine, immunoreactive trypsinogen

## Abstract

Are free carnitine concentrations on newborn screening (NBS) 48–72 h after birth lower in patients who develop type 1 diabetes than in controls? A retrospective case-control study of patients with type 1 diabetes was conducted. NBS results of patients from a Sydney hospital were compared against matched controls from the same hospital (1:5). Multiple imputation was performed for estimating missing data (gestational age) using gender and birthweight. Conditional logistic regression was used to control for confounding and to generate parameter estimates (α = 0.05). The Hommel approach was used for post-hoc analyses. Results are reported as medians and interquartile ranges. A total of 159 patients were eligible (80 females). Antibodies were detectable in 86. Median age at diagnosis was 8 years. Free carnitine concentrations were lower in patients than controls (25.50 µmol/L;18.98–33.61 vs. 27.26; 21.22–34.86 respectively) (*p* = 0.018). Immunoreactive trypsinogen was higher in this group (20.24 µg/L;16.15–29–52 vs. 18.71; 13.96–26.92) (*p* = 0.045), which did not persist in the post-hoc analysis. Carnitine levels are lower and immunoreactive trypsinogen might be higher, within 2–3 days of birth and years before development of type 1 diabetes as compared to controls, although the differences were well within reference ranges and provide insight into the pathogenesis into neonatal onset of type 1 diabetes development rather than use as a diagnostic tool. Given trypsinogen’s use for evaluation of new-onset type 1 diabetes, larger studies are warranted.

## 1. Introduction

Studies have demonstrated metabolic perturbations in patients who develop type 1 diabetes, even prior to onset of demonstrable antibodies [1,2]. These studies, while robust, utilized cord blood, which reflects placental and maternal admixture and may not be accurate for certain metabolic intermediates [3]. Newborn screening (NBS) is an extremely successful global public health initiative that screens for rare inborn errors of metabolism (IEM) soon after birth (48–72 h in New South Wales (NSW), Australia) using dried blood spots (DBS) taken via heel-prick. The immediate neonatal period is a time of intense catabolic activity where the neonate is adjusting from a continuous nutritional supply delivered via the placenta to intermittent feeding, with use of brown fat stores in-between periods of feeding making it an ideal period to test for metabolic intermediates of fatty acid oxidation disorders (FAOD), such as short-, medium-, and long-chain acylcarnitines [4], some of which have been associated with diabetes development [5,6,7,8,9]. Multiple studies have demonstrated the utility of analysis of pre-existing NBS results in various neonatal conditions [10,11,12,13,14,15,16], including diabetes [17,18,19,20,21,22,23]. Only one study has demonstrated differences in NBS results in patients who developed type 1 diabetes [23]. We sought to determine further detectable differences in NBS analytes in our cohort of patients with type 1 diabetes.

## 2. Materials and Methods

### 2.1. Study Population

This was a retrospective study undertaken at Campbelltown Hospital in Sydney, Australia. NBS results of patients with type 1 diabetes born after 1 April, 1998 (the year mass spectrometry was included for routine NBS) of the Macarthur Diabetes and Endocrine Services were requested from the NSW NBS Programme (NSWNBSP). NBS samples were obtained via heel-prick, between 48 and 72 h after delivery as part of routine clinical care. Blood was blotted into newborn screening collection paper, air-dried and mailed to the NSW Newborn Screening Programme. Conditions screened in Australia are primary congenital hypothyroidism, cystic fibrosis, galactosemia, and over 40 inborn errors of metabolism of amino acids, organic acids, and fatty acids. The analyte panel includes thyrotropin (TSH), immunoreactive trypsinogen (IRT; the immunoassay technique used does not differentiate between trypsin, trypsinogen, and trypsin- α-1 antitrypsin complexes), galactose, 6 amino acids, free carnitine, and 17 acyl-carnitines. Free carnitine (both a primary marker and a marker of degradation of acylcarnitines) and its esters alter quicker in a high temperature, high humidity environment post collection, which is why transport to the laboratory is kept to a minimum and all filter paper cards are analysed on day of receipt. After use, all cards are stored in accordance with Ministry of Health policy and discarded after 18 years. For disorders detectable by tandem mass spectrometry, a shift from the derivatized to underivatized sample preparation was implemented in 2007. The change to underivatized methods did not significantly affect the expected proportions of carnitines and their esters.

Pragmatic ratio of 1:5 controls for the same work date, batch run and hospital of birth, weight, sex, and gestational age as much as possible was used to minimize analytical measurements of uncertainty. Data concerning labour, delivery, and ethnicity were not available on the NBS samples.

This project received ethics approval (HREC reference: LNR/18/LPOOL/40) from Sydney Southwest Research Ethics Committee.

### 2.2. Statistical Analysis

Categorical data were compared using chi-square testing. It was decided a priori that the primary outcome would be free carnitine concentrations (CAR) as this had previously been demonstrated [23]. A secondary analysis of 22 analytes to explore other associations would then be performed (See Appendix A). Multiple imputation was used to account for missing gestational age for 12 cases (8%). Data augmentation algorithm in Stata (V16) MI procedure was used to generate 30 imputed data sets [24]. Parameter estimation using conditional logistic regression on each filled-in data set was performed to evaluate the relationship between type 1 diabetes development and each analyte, adjusting for gestational age. Rubin’s formulae were used to combine the parameter estimates and standard errors into a single set of results [25]. Hommel’s step-up technique was used for post-hoc analysis again using Stata (V16).

Newborn screening analytes are not normally distributed and reported as medians. Analytes added, as the program progressed, to improve sensitivity and predictive values were included in the analysis. Alanine is no longer part of routine NBS in Australia due to lack of clinical utility.

## 3. Results

NBS results were available for 159 patients and 696 controls (54 cases were not matched 1:5). Mean gestational age was 250 vs. 274 days respectively (*p* ≤ 0.001); mean birth weight was 3.4 kg overall (*p* = 0.66). Median age of diagnosis was 8 [1,2,3,4,5,6,7,8,9,10,11,12,13,14,15,16] years. Eighty (50.3%) probands were female; 86 (54.1%) were antibody positive. Fourteen had a family history of type 1 diabetes. Fifteen had concomitant immune disorders.

Free carnitine concentrations were significantly lower in those that went on to develop type 1 diabetes as compared to normal controls (Table 1). These changes were well within reference ranges and unlikely to be deemed pathological. A secondary analysis of 22 other analytes demonstrated a statistically significant association between higher concentrations of IRT and the development of type 1 diabetes compared to controls (Table 2). This association was no longer significant after adjusting for multiple analysis. Both medians were within the reference range, although a disproportionately higher number of infants in the type 1 group (8/159; 5%) had IRT levels that triggered second-tier testing.

## 4. Discussion

It is widely accepted that type 1 diabetes is autoimmune in nature, but the metabolic networks that precede antibody development, and their role in autoimmunity, remain poorly understood. Most metabolomic studies in type 1 diabetes utilize cord blood, which is reflective of placental/maternal metabolism. Infants once born no longer receive a steady nutrient source and rely on fat and glycogen stores prior to feeding. NBS samples, collected 48–72 h after delivery via heel-prick give a closer approximation of neonatal metabolism in a catabolic state, which is optimal for detecting FAOD. The only published study on NBS and type 1 diabetes development before 6 years of age found significantly lower median levels of free carnitines, and lower total mean carnitine, acylcarnitine fractions and lower total amino acids. Our cohort also had lower free carnitine levels, but no significant differences in acyl carnitines or amino acid levels as compared to the controls. An important caveat is that the differences were very subtle, did not trigger second tier testing, and were well below concentrations that could be considered pathological. While the primary purpose of NBS remains that of screening for rare inborn errors of metabolism, our work has demonstrated that results from NBS may provide insight into disease pathophysiology within known metabolic networks.

L-carnitine is well known for its role in fatty acid transport into mitochondria for beta-oxidation. It also improves skeletal muscle glucose uptake promoting glucose use instead of storage [26]. Immunomodulatory, anti-apoptotic, and anti-inflammatory effects of carnitine are known [27], and the role of acetylcarnitine (C2) in thymic regulation has been hypothesized [23]. L-carnitine has been shown to lower markers of inflammation in chronic disease [28]. An argument against causality of low carnitine and autoimmunity are patients with primary carnitine deficiency (OMIM 212140) where mutations in carnitine transporters (OCTN2) cause renal carnitine wasting. A higher incidence of autoimmune disorders has not been described in these patients [29], although they are treated with carnitine at diagnosis (through NBS). Promoting renal excretion of metabolic by-products during catabolism is another possibility, given similarities in metabolic crises between type 1 diabetes and organic acidurias (hyperglycaemia and ketoacidosis, with different ketoacids produced depending on the metabolic block). The immediate neonatal period is a crucial time of maturation and growth of β-cells, and another possibility is that lower values of carnitine either may influence this process or may be a marker of disturbed growth. Low carnitine levels have previously been demonstrated in patients already diagnosed with type 1 diabetes [30], but this was attributed to increased urinary clearance of organic acids. The precise reason for lower free carnitine levels soon after birth in our patients with type 1 diabetes at birth compared to controls remain unclear.

Our results also demonstrate a trend towards significance of elevated immunoreactive trypsinogen as a group in patients who developed type 1 diabetes, with ~2.5 times as many of these patient’s results triggering second-tier testing over the general population. Routine protocol for IRT second-tier testing in NSW is inclusion of 2% of the population tested, versus 5% in our type 1 cohort. Trypsinogen, the zymogen of trypsin, is secreted by the pancreatic acinar ducts (exocrine pancreas), and rises to actionable levels in perinatal stress, intercurrent illness, and cystic fibrosis (CF) carrier infants [31]. There is a physiologic rise in trypsin in the first 24 h after birth on transition from placental to enteral nutrition, with levels gradually normalizing.

While the exocrine and endocrine components of the pancreas are conventionally treated as separate compartments within the same organ, recent data have demonstrated lower levels of trypsinogen as a marker in antibody positive, newly diagnosed patients with type 1 diabetes [32]. Trypsinogen is known to be reduced once the diagnosis of type 1 diabetes has been established [33]. The trend towards statistical significance of raised IRT in our cohort warrants further study with a larger sample size. It is possible that, much like CF where raised IRT heralds pathology, raised trypsinogen levels are detectable soon after birth in patients who develop type 1 diabetes as compared to controls. This may be a subtle sign of pancreatic stress that is apparent in the immediate neonatal period when fat and glycogen stores are mobilized prior to overt beta cell dysfunction.

Our study has several limitations. All the cases are from a single centre although they were born in different hospitals throughout New South Wales. The sample size is relatively small and not all cases had antibodies available. The sample size was also too small to assess any differences across type 1 diabetes endotypes [34]. Some cases had missing data (gestational age). Nevertheless, the methods used to correct confounding for missing data and the statistical methods used minimized the possibility of a type 1 error. An important strength of our study is the timing of sampling. Most studies that have evaluated the metabolome of patients who developed type 1 diabetes used cord blood, which may be influenced by maternal and placental analytes that may act as confounders. In contrast, the timing of sampling in NBS (48–72 h in NSW, Australia) minimizes maternal and placental influence on analyte measurements and provides information on the neonatal metabolic milieu as it adjusts to life ex-utero. Analytic measurements were performed by only one laboratory (NSW Newborn Screening Program) minimizing systematic bias associated with varying measurements. Controls were matched to account for minimizing measurements of analytic uncertainty, again minimizing potential for result variation. We also had a carefully phenotyped group of patients whose ages of diagnosis varied (in contrast to the only other study done where the age of diagnosis was 6 years and below). Importantly, being able to evaluate these biomarkers early in life, prior to any other environmental exposures that may influence the development of type 1 diabetes enhances the causal inference of this study’s conclusions.

## 5. Conclusions

In summary, patients who developed type 1 diabetes demonstrated lower free carnitine concentrations. There was a trend towards statistical significance for IRT. These concentrations may show perturbations soon after birth in patients who eventually develop type 1 diabetes. The changes were well within normal ranges, did not trigger the need for confirmatory testing (although IRT levels among cases had higher frequencies of second-tier testing) and lays important groundwork for furthering understanding regarding type 1 diabetes development.

## Figures and Tables

**Table 1 nutrients-13-03669-t001:** Characteristics of cohort.

	Type 1 *N* = 159	Controls *N* = 696	*p* Value
Demographics			
Gender	M: 80 (50.3%) F: 79 (49.7%)	M: 341 (49%) F: 355 (51%)	0.95
Gestational age (days)	273(259–280)	280(272–280)	0.00
Birth weight (kg)	3.36 (3.06–3.75)	3.43(3.16–3.72)	0.66
Age at diagnosis(median) Antibody status Family history Other immune disorders	8 years (1–16 years) 86/159 (54%) 14/159 (9%) 15/159 (9%)	- -	

- missing values.

**Table 2 nutrients-13-03669-t002:** Results of primary analysis.

NBS Analytes (Unit/Whole Blood)	Type 1 *N* = 159	Controls *N* = 696	Coefficient	T Statistic	*p* > |t	95% Confidence Interval
Free carnitine (μmol/L)	25.50 (18.98–33.61)	27.26 (21.22–34.86)	0.009	−2.37	0.018	−0.041 to −0.004
IRT (µg/L) *	20.24 (16.15–29-52)	18.71 (13.96–26.92)	0.012	2.01	0.045 *	0.001 to 0.023

* This value did not reach statistical significance following Hommel step-up adjustments.

## Data Availability

Data will be made available upon request.

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
