# Peer review of "Immunoreactive Trypsinogen and Free Carnitine Changes on Newborn Screening after Birth in Patients Who Develop Type 1 Diabetes"

_nutrients, 2021, doi:10.3390/nu13103669_

Round 1

Reviewer 1 Report

This is a well written, interesting manuscript, employing appropriate methodology that will contribute to the literature.

Author Response

Thank you for the comments!

Reviewer 2 Report

Overall, the manuscript by Estrella, et al., provides an interesting and novel exploration of potential NBS biomarkers for type I diabetes. The manuscript is clearly written and easy to follow and provides a novel analysis. The below suggestions could be incorporated to enhance the paper:

Line 46: Odd period within sentence with seemingly missing words. 

Materials and Methods: A brief discussion around stability of free carnitine and other associated analytes would be warranted along with a description of the storage conditions of the cards (e.g., temperature, dessicant use, etc). This would help explain why or why not any corrections for sample decay (and in the case of free carnitine, likely increase given acylcarnitine hydrolysis) was performed. The use of matched controls may have mitigated this need, but additional description would be helpful.

Line 76: Italicize a priori

Line 93: Missing period after (54 cases were not matched 1:5).

Results: It is mentioned in 2.1 that there was a switch from derivatized to underivatized methods in 2007. Was there any effect on free carnitine levels between patients and controls before 2007 and after? 

May also consider examining race of patients and controls, especially as it pertains to IRT as it is has been shown that Black newborns have higher IRT levels (I am not overly familiar with the racial make-up of Sydney, so this may be a non-issue).

Author Response

Overall, the manuscript by Estrella, et al., provides an interesting and novel exploration of potential NBS biomarkers for type I diabetes. The manuscript is clearly written and easy to follow and provides a novel analysis. The below suggestions could be incorporated to enhance the paper:

Thank you so much for the comments.

Line 46: Odd period within sentence with seemingly missing words. 

Thank you for picking this up. We have deleted the double-written paragraph which must have occurred at the time of transferring the manuscript to the template.

Materials and Methods: A brief discussion around stability of free carnitine and other associated analytes would be warranted along with a description of the storage conditions of the cards (e.g., temperature, dessicant use, etc). This would help explain why or why not any corrections for sample decay (and in the case of free carnitine, likely increase given acylcarnitine hydrolysis) was performed. The use of matched controls may have mitigated this need, but additional description would be helpful.

Thank you for this comment. While newborn screening cards are kept in a controlled environment between time of collection and dispatch to the centralized laboratory as well as at the laboratory, there is less control during transport. Transport is therefore kept to a minimum with samples from Campbelltown received on the day of dispatch. Further modification of results was avoided as we did not perform new analysis or re-analyze any of the existing cards. The analysis done as part of our methodology utilized pre-existing results that have been stored on the NSWNSP database. Carnitine and its esters may degrade quicker in transit when temperatures are extreme and in high-humidity situations however as indicated the matched controls mitigated this need when assessing trends between groups

              We have amended to sentence to read: ”Free carnitine (both a primary marker and a marker of degradation of acylcarnitines) and its esters alter quicker in a high temperature, high humidity environment post collection which is why transport to the laboratory is kept to a minimum and all filter paper cards are analysed on day of receipt; after use, all cards are stored in accordance with Ministry of Health policy and discarded after 18 years. (Line 66-68)

Line 76: Italicize a priori. 

Thank you. This has been corrected.

Line 93: Missing period after (54 cases were not matched 1:5). 

Thank you. This has been corrected.

Results: It is mentioned in 2.1 that there was a switch from derivatized to underivatized methods in 2007. Was there any effect on free carnitine levels between patients and controls before 2007 and after? 

Thank you. There was a change in free carnitine levels between cases and controls, which was expected and proportionate to the change in methodology.

We have added a line to say “the change to underivatized methods did not significantly affect the expected proportions of carnitines and their esters.” (Line 71-73)

May also consider examining race of patients and controls, especially as it pertains to IRT as it is has been shown that Black newborns have higher IRT levels (I am not overly familiar with the racial make-up of Sydney, so this may be a non-issue).

Thank you. Sydney is multicultural with a mix from all ethnic groups. Whilst ethnicity data is not collected for initial newborn screening analysis to ensure equity of screening, ethnicity data is collected for cases. None of the cases are of black ethnicity. Blacks form a small minority of the population, with 168,000 people identifying as Black African Australians by birth in 2019.